# Understanding Social Impact and Value Creation in Hybrid Organizations: The Case of Italian Civil Service

**Paolo Esposito** [1], **Valerio Brescia** [2,*], **Chiara Fantauzzi** [3] **and Rocco Frondizi** [3]

1   Department of Law Economics Management and Quantitative Methods, University of Sannio di Benevento, 82100 Benevento, Italy; pesposito@unisannio.it
2   Department of Management, University of Turin, 10134 Turin, Italy
3   Department of Management and Law, University of Rome Tor Vergata, 00133 Rome, Italy; chiara.fantauzzi@uniroma2.it (C.F.); rocco.frondizi@uniroma2.it (R.F.)
*   Correspondence: valerio.brescia@unito.it

**Abstract:** The aim of this paper is twofold: first, it aims to analyze what kind of value is generated by hybrid organizations and how; second, it aims to understand the role of social impact assessment (SIA) in the measurement of added value, especially in terms of social and economic change generated by hybrids. Hybrid organizations are a debated topic in literature and have different strengths in responding to needs, mainly in the public interest. Nevertheless, there are not many studies that identify the impact and change generated by these organizations. After highlighting the gap in the literature, the study proposes an innovative approach that combines SIA, interview, interventionist approach and documental analysis. The breakdown of SIA through the five elements of the value chain (inputs, activities, outputs, outcomes, and impact) guarantees a linear definition of the value generated through change with procedural objectivity capable of grasping hybrid organizations' complexity. The value generated or absorbed is the change generated by the impact measured based on the incidence of public resources allocated. Through the SIA and counterfactual approach, the civil service case study analysis highlights how the value generated by public resources can be measured or more clearly displayed in the measurement process itself.

**Keywords:** hybrid organization; social impact; value creation; civil service; social impact assessment

## 1. Introduction

Academics have been debating the definition and outcome of hybrid organizations [1]. In the literature, there are three classic forms of hybrid organizations: hybrid public companies formed by subsidiaries and investee companies, social cooperatives to which the public provides a mandate for achieving specific objectives and social enterprises [2,3]. However, the classical forms of hybrid organization are added to other more complex forms and have not yet been studied in the literature. An example is given by those third sector organizations that receive public funds, where in fact, the public sector is a partner in specific activities and in some cases also involves the private sector to achieve shared social objectives. In these cases, the interest in the project's impact and the value created become recurring themes of interest for the new organizational forms created. The subjects involved in the hybrid organization pursue the common interest and support the achievement of social, environmental, or economic needs by overcoming potential obstacles thanks to the characteristics of each one who in such organizations unite to form a new organizational form [4]. The reform linked to New Public Management has led to different types of outsourcing of public interest services, some typically oriented towards new corporate forms under public control, others towards the public–private partnership, others towards new hybrid solutions in which different organizational forms are united to answer the need [5–8]. Furthermore, since 2017 in Italy, the country has

witnessed the reorganization of legislation on the third sector, including associations (social promotion association, voluntary associations), social cooperatives, social enterprises, ecclesiastical bodies, and foundations [9]. The new legislation increasingly directs existing organizations towards general interests of public interest, with objectives of social and environmental sustainability as already highlighted for non-profits and the possibility of carrying out commercial activities with a type of taxation different from those profit companies community-oriented shared public purposes [10,11]. In several cases, the reform has changed the third and non-profit, making it part of the public fabric through new forms of hybrid organization to respond to a lack of available resources and economic difficulties [10,12,13]. The recent global financial crisis and the pandemic triggered by COVID-19 have had various effects on evaluating public performances that have led to budget cuts and the search for new organizational forms to meet the need with fewer resources available [14,15]. To decrease the available public budget associated with better systems of efficiency and use of public resources, the parallel stream of research on sustainable business models capable of responding to the economic, social, and environmental system's developmental complexity has been added [16–18]. The approach provided by hybrid organizations could support the correct use of resources, the limited use of the same response to a paradigm already studied by O'Flynn [19], who questioned himself with the adoption of new public management in the twentieth century of managerial approaches borrowed from the private sector and applied to the public sector to increase public value. The public value generated by the adoption of new managerial approaches, as clarified by Stoker [20], includes the involvement of stakeholders in the collaborative and production process as a way of overcoming the previous model. From this perspective, the analysis conducted sees the subjects traditionally involved as stakeholders on equal components of hybrid organizations.

O'Flynn [19] identifies in the concept of public value a new approach that adheres to that defined by hybrid organizations which, as theorized by the scholar, have a greater capacity for collecting preferences, a multi-accountability approach that involves all subjects, the ability to pursue multiple objectives including service results, satisfaction, results, trust, and legitimacy. The public value generated by hybrid organizations can be determined through the social and economic impact of the projects and services of public interest carried out [16,21]. Public value offers a broader way of measuring government performance and guiding policy decisions, according to Kelly, Mulgan, and Muers [22], where public value could measure the impact of public interest projects. The concept of performance recalled in the paper therefore refers to the holistic conception of public value. The literature on hybrid organizations has not yet defined the public value it can generate either through a meaningful approach or case studies. The analysis conducted focuses on an example of a hybrid organization generated by a large project shared between the different types of organizations to merge them into a single organization with the same expectations and interests. This is the Italian national civil service. The theoretical approach proposed to define public value in hybrid organizations can be generalized as it responds to the verifications and requirements identified by Ruddin [23], who also highlights how an approach that can be generalized to other case studies has practical relevance in social studies. In the analysis conducted, the same hybrid organization approach applied to civil service projects can be adopted in different European contexts. The analysis conducted after a brief description of the existing literature seeks to identify a possible approach to determine the hybrid organization's social impact and value [24]. The approach to facilitate understanding of the logical passage divides the project into the five phases of the social impact assessment approach [25,26]. Interviews, collection of results, interventionist approach, and application proposal for evaluating the added value allow the inductive passage [27] from practice to theory and vice versa, highlighting and understanding the main aspects. The study initiates a process of reflection on complex hybrid organizations and the possibility of measuring the impact generated through an innovative approach from a logical point of view. Following the research gap and the observation of empirical

phenomena not explained by existing studies, our analysis will provide answers to the following research question (RQ):

RQ1: How do hybrid organizations generate value and what kind of value is it?
RQ2: How does the social impact assessment measure the phenomenon of value creation linked to change?
RQ3: Is social impact assessment a useful tool to understand the use of public resources generated by hybrid organizations?

The study consists first of an analysis of the literature and the second section presents the methodology and the inductive approach on the specific case divided into phases and recalls through the social impact analysis. The findings guide the reader in the subsequent discussion and conclusions, providing practical elements to demonstrate the approach's capacity and generated value. The discussion guides the reader to understanding social impact. The conclusion highlights the international debate theme and the possible application of the results in an international context.

## 2. Theoretical Framework

This section is dedicated to the theoretical framework of the paper and it is composed of two main pillars. First of all, the purpose is to analyze several definitions of hybrid organizations. Then, the aim is to analyze the academic debate on impact measurement in hybrid organizations, individuating a literature gap on value creation evaluating to fill. The debate on the issue of social impact is generated by the consideration of multiple criteria of different nature (economic, environmental, and social), as well as the transparency and engagement of the different stakeholders, such as organizations, government, and communities oriented towards mapping resource sustainability in a complex environment.

### 2.1. Framework

The term "hybrid organization" is used to combine elements from both for-profit and non-profit sectors, to maintain a mixture of market and mission oriented practices, and to address economic, social, and ecological issues [28,29]. Nevertheless, there are different definitions of hybrid organizations, due to their nature and complexity [30].

By definition, hybrids are the offspring of different species [31] and, for what concerns the organization and management literature, the term has been employed to describe organizations that draw on at least two different paradigms, logics, and value systems, allowing the rise of a new conception of economic organizing [32]. In this sense, hybrid organizations can be seen as a new form of organization, able to compete not only on the quality of goods, but also on the capacity to effect positive social and environmental change [33]. Boyd et al. [34] defined hybrid organizations as entities that are market-oriented and mission-centered, which can be studied on the basis of two specific criteria [35]:

- they have a business model aimed to create social value;
- they are able to generate income to sustain their operations.

Hybridity represents a mixture of several heterogeneous components and does not refer to something new, but to a new combination of existing elements. Besharov and Smith [36] used this term to indicate the coexistence of two or more distinct forms of organization. Jay [37] provided a similar definition, according to which hybrid organizations are seen as entities able to combine multiple institutional logics to solve complex problems. Indeed, a hybrid organization is driven by two forces, represented by social change and the sustainability of organization [38], offering a blended value proposition, composed of economic, social, and environmental components. It is in this sense that, according to Santos [39], this type of organization is mainly focused on creating value rather than capturing it, while Haigh and Hoffman [40] underline its ability to provide high quality differentiated goods and to pursue both a social and environmental mission. Furthermore, Boyd et al. [34] state that hybrid organizations are generally characterized

by a long-term perspective on profit and a very close and personal relationship with their crucial stakeholders (suppliers, producers, and customers).

According to Grossi et al. [41], in the field of public administration and management, hybridity refers to organizations composed of structural elements deriving from other types of organizations belonging to different sectors (private, for profit, and third sector). It is within such triangulation that they identify specific hybrids:

-   public/private for profit hybrids, like state-owned enterprises;
-   private for profit/third sector hybrids, like social enterprises or co-operatives;
-   public/third sector hybrids, like welfare associations or other organizations sponsored by government.

The concept of hybrid organizations has been adopted to describe various configurations of cross-sectoral collaboration, such as network and hierarchy [42], government and business [43], academic and market [44], healthcare and science [45]. According to Doherty et al. [46], by spanning the boundaries of private, public, and non-profit sectors, hybrid organizations are able to bridge institutional fields, facing conflicting institutional logics. In this sense, they are able to earn trust and the authority to establish connection and dialogue between several categories of actors, also by including former opponents [47].

Given these considerations, for the purpose of this paper, we adopt the definition provided by Jolik and Niesten [48], according to which hybrids represent collaborations between independent organizations that exchange and develop goods and services to create value, reduce agency and transaction costs and allocate residual claims, by combining resources, organizing information, and safeguarding contractual hazards and property rights. Despite the recent increased interest in hybrid organizations, the literature appears fragmented across many academic disciplines over several decades.

According to one of the most popular approach, hybrid organizations are seen as a continuum between sectors [49,50], while other authors preferred to follow a "single sector emphasis" [51–54], studying hybrid organizations from the perspectives of one specific sector, the public or the private one.

Other writers have gone further in a separate sector approach, looking at hybridization and hybrid organizations as the permanent features in the welfare system [55,56].

Furthermore, numerous studies were carried out with the aim to individuate the factors on which depends the likelihood to develop a hybrid organization [57]. They focused on the motivation of traditional entrepreneurs. Some showed the desirability for self-employment, tolerance for risk and self-efficacy at the center of their interests [48]. The literature on hybrid organizations presents hybridity as an enabling condition to achieve legitimacy from different institutional logics, in order to survive [28,58]. On the other hand, organizational stakeholders and policy-makers can become "institutionally confused" if an hybrid's behavior does not match the description of the ideal-typical form of organization in contemporary society [59]. In this sense, even if hybrid organizations have to solve tensions regarding their identity, according to Battilana and Lee [60] and Santos et al. [39], the conflicting logics they respond to and the practices they implement can be seen as the essence of their constitution, the condition to handle specific and complex situations [61].

The specific concept of hybridity in public administrations still appears undeveloped [62]. Several studies have analyzed hybridization within public organizations [63], considering structural and cultural complexity as its main feature [64,65]. Structural complexity in public organizations can be measured in terms of vertical specialization and horizontal specialization, both characterized by intra and inter-organizational elements [58–60]. Vertical intra-organizational specialization indicates how formal authority is distributed among different levels of hierarchy. Vertical inter-organizational specialization instead focuses on specialization among public organizations (ministries with many subordinate agencies). On the other hand, horizontal intra-organizational specialization indicates the internal specialization of public organizations, while horizontal inter-organizational focuses on the level of specialization among public organizations at the same levels. Cultural complexity, instead, shows the variety of informal, cultural norms and value within

and among public organizations. A weak level of cultural complexity means cultural homogeneity and integration, with members who are all committed to the same norms and values. On the basis of these considerations, Christensen and Laegreid [63] affirmed that hybridization in public organizations can assume different meanings, which can be explained through complexity (in all its configurations). More deeply it reflects potential inconsistency between diverse structural and cultural elements in a public entity. In this conception, hybrid structures that follow different organizational principles can link new means of coordination and traditional sectoral agreements and be an effective way to manage the "coordination paradox" (i.e., vertical coordination measures can counteract horizontal coordination). However, the performance and effects of these practices are often conflicting and uncertain, and there is a trade-off between potential gains through flexibility and disadvantages through ambiguities, tensions and conflicts that fall within the sphere of cultural elements. Coordination is a crucial structural element for governance and quality capability because it shapes program design and influences efficiency gains, which in turn affect government legitimacy.

### 2.2. Hybrid Organizations and the Determination of the Generated Value

Academics and scholars are now debating the fallout that different organizational forms have in terms of impact measurement [66] based on environment, workers, community, and governance and managerial approaches and frameworks adopted for measuring the previous elements. The economic aspect actually has an impact on the reference community and internal elements of the organization in the hybrid organization model [66]. If different approaches have been identified to map and measure the impact of organizations whose boundaries are well defined, the literature cannot define numerous application cases and possible approaches of objective measurement within hybrid organizations [1,67]. The definitions provided also identify social cooperatives and hybrid organizations in the literature as typical examples of hybrid organizations in which, at the same time, interests oriented to generate profit from the private sector, respond to community needs as required by the public sector, and achieve social objectives or environmental as required mainly by non-profit organizations [46,68]. The two forms were the first to be fully configured as hybrid organizations. The third sector is increasingly configured as a hybrid organization withouthaving identified appropriate tools for measuring and evaluating the value generated by the impact of the activities often organized in collaboration with the public and private sectors. Several articles analyze how collaboration or partnership with the private sector can generate value [69,70] or destroy value [71]. Several articles suggest that the partnership between public and third sectors can generate value in the community context [72]. To the studies, there are also methodological proposals for mapping the social impact in the third sector [73,74]. However, there are no studies on how to determine the impact of hybrid organizations objectively [28,46]. The two studies highlighted the absence in the existing literature reviews of a common approach that considers mission drift and challenges to legitimacy in hybrid organizations. They recalled particular social enterprises as a classic example of hybrid organization [41] and the difficulty in defining the value generated by hybrid organizations—public and third sector in the definition of value for the territory and the reference community [23]. Hybrid organizations can generate excellent value, especially when welfare is transforming into a welfare mix in which the public needs the private sector and the third sector to achieve efficient and effective results in the social field under the principles of New Public Governance and New Public Management [75]. Moreover, if the public defines the objectives of general interest, approves the activities, finances them, or co-finances them and requires the collaboration or prevalent implementation by the third sector and the private sector, how can this generated value be measured? Esposito and Dicorato [76] highlight how the public sector needs to determine the impact of the resources used through the partners' ability to generate value, legitimize social activities, and guarantee operational and financial sustainability. The literature gap linked to both hybrid organizations and the evaluation of the value

generated is repeated and coincides with a mutual interest in enhancing the impact on the social and economic system.

For what concerns the partners' ability to generate value, according to Rhodes [77] and Osborne [78], this depends on the ability to build partnerships and inter-actions between different sectors (generally the public and the private ones), promoting cooperation as well as equity and democracy. Nevertheless, different approaches for what concerns the measurement of public value can be retraced overtime [79]. The public value generated by hybrid organizations can be determined through the social and economic impact [11,16,74]. Indeed, if in line with the Bureaucratic Public Administration (BPA) public value was evaluated in terms of legitimacy and formal correctness [80], the New Public Management focused on its economic dimension [81]. On the other hand, the theories of network governance stressed the role of interdependencies for multi-actor collaboration and how meta-governance can manage and provide direction to such networks [82–84].

In this sense, co-production assumes a significant relevance for public value production and engages citizens not only in voting [85], but also implies a redistribution of power between professionals and citizens, pointing out important issues related to accountability [86]. In order to co-produce public value, different actors have to share the same perception of public value and then they have to develop specific capacities such as being able to overcome significant gaps, divergent resourcing, and differing time frames. Emerging literature has identified different mechanisms to manage potential conflicts between actors at different levels [87–89] and, among these, hybridization appears as the crucial one by sustaining distinct policies and practices that pursue competing values.

Kelly et al. [22] studied public value on the basis of its three "building blocks," represented by services, outcomes and trust. According to such an approach, the aim is to generate outcomes in society by providing public value through services. Furthermore, other authors have started to focus on new and intangible aspects, such as citizen participation and the need to satisfy their demand and overcoming the previous public sector paradigms [90–92]. This new attention can be seen as the main contribution provided by the Public Value theory to evaluate the value generated by hybrid organizations.

Lastly, even with the aim to measure performance, Faulkner and Kaufman [93] carried out a model based on four different dimensions which can be adapted to several contexts, including hybrid organizations. The first dimension is related to social, economic, cultural, and environmental outcomes [4], while the second one is about trust and legitimacy, since public activity must be legitimized by all stakeholders involved, stimulating trust [91,92,94]. The third dimension takes into consideration the quality of public service, which is a crucial aspect in the provision of public services and involves satisfying users' needs [95–99]. The latter is dedicated to efficiency, with the aim to minimize costs [100], bureaucracy [101], and value for money [102].

As anticipated, a second current of theories tends to evaluate the impact of social activity through social legitimacy [76], highlighting the importance of specific factors such as autonomy, trust, steering, and ability to influence performance [103].

Performance represents a complex concept to define, especially for what concerns hybrid organizations. While for-profit organizations measure it in terms of outputs—taking into account their effectiveness, efficiency, and productiveness—for non-profits it is harder to quantify goals, since they operate in less competitive environments which makes benchmarking more difficult [104,105].

From a literary point of view, there are different opinions for what concerns the effect of an organizations' autonomy on performance. I On the one hand, autonomy can create transaction costs which negatively affect performance [106,107]. On the other hand, autonomy can be beneficial, valuing citizens' satisfaction rather than efficiency. According to Cambini et al. [108] and Swarts and Warner [109], autonomization ensures cost savings, but it depends on the sector where it takes place [110].

Generally, strategic steering is associated with positive effects on performance, since it can help communicate what level of performance is expected, to assess whether objectives

are achieved and to align management's interest with that of governance establishing sanctions and rewards [111,112]. But, according to Benabou and Tirole [113], extrinsic motivation through incentives can replace intrinsic motivation, generating distrust.

The last approach that can help in assessing hybrid organizations' performance suggests looking at the concept of sustainability, considering not only the social and environmental impact related to activities but also wider issues such as climate change, social stability, job creation, and the protection of human life [114,115]. According to Mahadi and Sino [116], understanding the notion of public value represents the first step to achieve sustainable development and deliver services satisfactorily. Sustainability represents a fundamental driver for public–private partnership activities, allowing the pursuit of public and private partners' goals without compromising the capacity to meet the needs of their reference communities. The same could be said for hybrid organizations, which overcome the risk of conflicting logics and ensure a positive social impact.

## 3. Method Approach

In Italy, the civil service represents a typical example of the transfer of public funds to cover general interest activities that involve users present throughout the national territory. The civil service has similar characteristics in other European countries. In Italy, civil service is voluntary and can be carried out as an alternative to military service as in France, Germany, Spain, Holland, Poland, Portugal, United Kingdom, the Czech Republic. In other countries such as Austria, Cyprus, Denmark, Finland, Greece, Latvia, Lithuania, Slovakia, Sweden, Switzerland, Norway, it is mandatory [117]. On 3 April 2017, the Civil Service in Italy was established with the characteristics currently in force according to the Italian Legislative decree 6 March 2017, n. 40. The Universal Civil Service, "is aimed at the non-violent and unarmed defense of the homeland and peace between peoples" and, "constitutes an institution of integration, inclusion and social cohesion, aimed at strengthening the relationship between the citizen and the institutions of Italian Republic, contributing to the stability of democratic institutions as well as to the construction of a participatory democracy and new forms of citizenship." This experience exclusively dedicated to young people between 18 and 28 years can be carried out in Italy or abroad, both in member countries of the European Union and others. Participation is bound to adhesion to a project, presented by a third sector, which can last from 8 to 12 months. The organization which organized the universal civil service is an excellent example of a hybrid organization as the selected projects have been taken advantage of and are co-financed by the Department for Youth Policies of the Italian State. The Department is partner and investor of the organization. The accredited body evaluates the impact obtained by the various projects in response to the Department's general objectives, and the offices that welcome the subjects involved are part of the third sector and non-profit. Therefore, there is an extension, control, and public interest that involves different organizational forms with non-uniform types of financing. The case study is subject to public contributions and public control. Having to respond to reporting purposes of a public nature and public accountability, it is itself a hybrid organization due to the principle of the prevalence of economic substance over legal form [118]. The case study considered is significant for the analysis that we intend to conduct because the Vol.To volunteer service center is one of the third sector bodies that welcomes and coordinates the largest number of people involved in the civil service. Vol.To was accredited as a National Civil Service Body in 2005 by enrolling in the Register of the Piedmont Region (Italy). Over time, the number of its reception centers increased to becomethe only private social body registered in the first class in the Regional Register (with several offices greater than 100) in 2017. With the reform of the third sector that establishes the Universal Civil Service, Vol.To obtained accreditation for 140 reception centers in collaboration with 72 organizations (non-profit and ecclesiastical centers) defined by the "Reception Bodies" standard.

The civil service project represents in its configuration a significant example of a hybrid organization where public interests, objectives, and projects are shared, approved, and

financed or co-financed by the public sector to generate social value. The bodies accredited to host the civil service subjects are either public bodies or third sector. The projects involve third sector partners, public or private, for the performance of specific activities falling within the objectives of general public interest. The empirical investigation and analysis of the generated value of the hybrid organization of the civic service analyzed, as well as the project realized by Vol.To, was supported by the Bank Foundation and established by law as a point of regional reference [119].

Analysis of the empirical data followed a systematic combining approach based on an abductive process [27] which is characterized by the interplay between rich longitudinal empirical data and literature. The longitudinal analysis of a case study allows, through different methodologies and approaches, identification and explanation of the same phenomenon with different sources and visions and responding to the gaps that a single approach provides in literature. Examples of longitudinal analysis of a case study are defined equally in the literature [120,121]. The project is defined in four phases:

(1) Analysis of the case study through Social Impact Assessment (SIA). SIA is the approach identified as a possible solution to mapping the process of hybrid organizations [25,26]. To analyze the approach, we refer to the phases of the value chain shown in the results section. The study recalls in the results the five phases for the impact assessment and the elements associated with each. These are input (human, financial and material resources), activities (transformation of inputs), outputs (products and services), outcomes (results and effects on beneficiaries) and phase impact (change in the reference community), which define the generated value [122,123].

(2) Creation of a chronological overview of the interview and interventions in the examined public value based on empirical data from documentation and interventionist workshops to structure a new approach to measure the impact and social change. The interventionist approach was conducted through five meetings which took place from three months before the start of the project up to one month after the end of the project. The interviews were carried out on subjects who participated in the civil service at the end of 2020 and onmanagers and employees of Vol.To between 2019 and 2020 to identify the best approach to be adopted for measurement. The interventionist approach involved the participation of one of the authors defining the tools, elements, and methodology to be adopted to measure social impact and value generated through an agreement between the University of Turin (Italy), Vol.To and the Presidency of the Italian Council of Ministers. The activity was developed in 2020. Interviews with volunteers, were developed by semi-structured questions aimed at defining the actual change provided by the project through the counterfactual method [124].

(3) Use of key concepts of the theoretical framework (SIA method) to link the empirical findings with measurement of impact in the hybrid organization. The data collected are available online. References were provided in open access in order to guarantee the replication of the result by the other researchers [125–127]. The determination of the social impact of the project described is part of an impact certification process by a European certifying third party. The approach is the derivation of a practical and theoretical training project which providedcertification of the skills on the social impact of the author who dealt with the interventionist analysis by the University of Turin (Italy) and a European skills certification body called CEPAS [128,129]. The approach on social impact assessment is one of those teachers and certified to one of the authors present in the SCH120 register held by CEPAS.

(4) Analysis of the overall results to answer the research question.

The impact is the actual assessment of the change and the value generated. Therefore, it will be described and commented on in the Discussion as a logical element of the approach's final determination.

The case study analyzes the fallout of the 2019 project of Vol.To completed in the year 2020.

Ninety-four people had access to the project, 36 of whom were males and 58 females, with 44 young people employed in Turin (Italy) and 48 in the Province of Turin. The selections rewarded graduates who have risen to 59, 6% (56 young people) and young people with low schooling were penalized, which fell to 13.8% (13 people). Graduates which slightly decreased, are 26.6% of the total number of operators who participated in Vol.To. Vol.To's civil service projects support the activities carried out by the third sector, which can strengthen their support for the reference communities with the help of volunteer operators involved. The needs and the projects activated identify 25.7% cultural animation activities for minors and young people, followed by initiatives aimed at users with disabilities (20%) and support for the elderly (17.1%). The other activities aim to support adults in distress, including women in difficult or poor health conditions, school tutoring, peace education, civil rights, and environmental protection activities. In Table 1 and Figure 1, all partners and direct stakeholders of the project interested in increasing their workforce or covering voluntary activities were already active with a transfer of values and skills.

**Table 1.** Hosting third sector organizations according to the classification defined by the Legislative Decree Italian 117/2017.

| Type of Organization | Percentage Distribution |
|---|---|
| APS (associations for social promotion) | 8.6% |
| ASD (amateur sports associations) | 2.9% |
| Association | 8.6% |
| Social cooperatives | 11.4% |
| Foundations | 8.6% |
| Odv (volunteer organizations) | 57.1% |
| Ecclesial bodies | 2.9% |

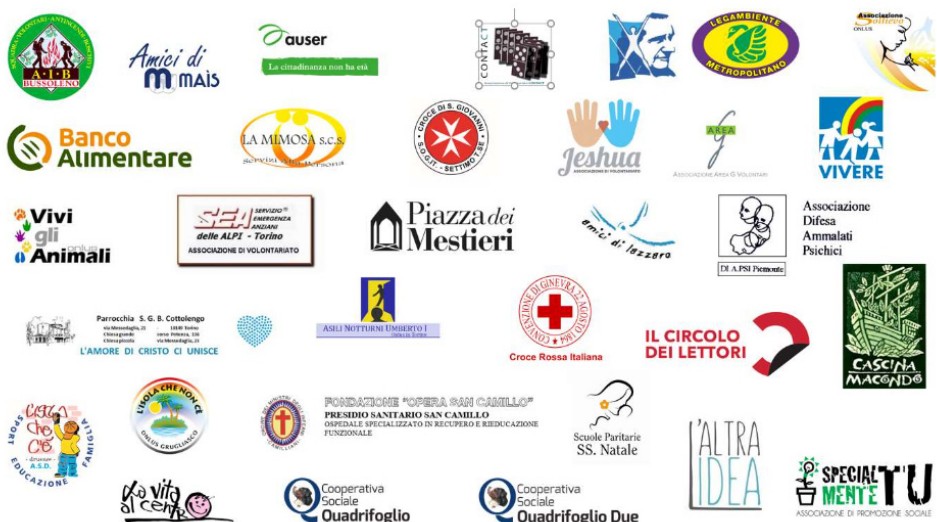

**Figure 1.** The host organizations of civil service Vol.To.

The representation of the impact of activities and stakeholders is made even more complicated as each third sector entity has one or more public, private or third sector entities as a partner (Appendix A).

## 4. Empirical Evidence

This section presents the interviews, the proposed approach for evaluation and enhancement of personnel, and the impact as conclusive elements of the SIA analysis in support of the study's empirical and theoretical tips.

## 4.1. Input

The project involved an overall investment of €42,911 against a ministerial loan of €7740. Three trainers were employed for general training: Rosanna Lopez: responsible for managing volunteers, Vol.To. Maida Caria: accredited trainer, Vol.To. Sandro Prandi: accredited trainer, Vol.To. Experts in general training were also employed concerning specific necessary skills not possessed among Vol.To employees: Nice Law Firm Stefano Lergo: civil protection expert Enrico Bussolino: third sector expert. They were involved after general training of trainers for specific projects and on average each project required six trainers, for a total of 185 trainers, of which 87 were male and 98 were female, with an average age of 51 years.

## 4.2. Activities

The activity carried out included acquiring generic skills on preordained topics and specific training provided within the host organizations with work activities in the project context defined for each. The generic training considered four macro-areas with a total duration of 42 h for each class (consisting of a maximum of 25 people each) delivered in the first 180 days to provide useful elements for understanding and elaborating the experience of the Civil Service. The macro-areas treated customized based on the project of interest concern the values and identity of the national civil service, active citizenship, and the young people within the civil service context. The frontal lesson (40%) is the classic technique for teaching, where the trainer deals with a specific topic using his/her studies and experiences in the field of civil service and related issues. Support of diversified instruments includes documentary screening, PowerPoint presentations, reading of texts, and testimonials from external experts. Non-formal dynamics (60%) includesthe most interactive techniques between the trainer and the group and between the members of the group itself such as role-play, simulations, plenary discussions, group work, expressiveness workshops, and sharing of personal experiences. Generic training supports the experience, although specific training and experience in the civil service are more characterizing and have a bigger impact. Each project required a period of specific training on project activities for a minimum of 50 h with teachers who are experts in the subject with many years of experience and/or a degree relevant to the activities envisaged. All the organizations involved chose to administer more hours than the minimum required. On average, the specific training lasted 75 h with a minimum of 72 h and a maximum of 108 h, delivered within the first 90 days of the project. On average, each project required six trainers, for a total of 185 trainers, of which 87 were male and 98 were female, with an average age of 51 years. It is sufficient to refer to the project activities indicated in the table below. In addition, each volunteer was accompanied by a tutor, who monitored and supported people (a minimum of 10 h per week). The need to which Vol.To responds through the project activities includes an activity equal to 30 h a week for 12 months. The areas of action concern minors and young people's cultural animation, organization and management of workshops for children, and participation in events for children and families (Figure 2).

The individual specific activities for each project falling within the defined activities of general interest can be found in Appendix B.

To be able to map the change in skills and knowledge of each subject, the civil service project envisaged participation in a skills certification path in collaboration with the University of Turin. The path started in parallel with training and field activities involving the subjects in five steps. Certification of skills is part of a project funded by the European Union and adopted for Erasmus+ courses' certification. The reference model used was born from a European LEVER UP project that was created in line with the Non-formal/Informal Learning Validation (NFILV), and Validation of Prior Learning (VPL) approaches. Creation of the reference model was created by partners between 2014 and 2016 in Italy, the Netherlands, Denmark, Spain, and Poland [130]. LEVER UP was created to help individuals enhance and make visible transversal skills and competencies acquired through non-formal and informal learning experiences, for example, by volunteering. This allows them to

increase their awareness, social responsibility, employability, and mobility. The first step is to create awareness of the value of the experience carried out in the volunteer/civil lawyer from a human and educational point of view. The key factor in this step recognizes that skills can be learned in informal settings, and volunteering is one of them. Each experience allows a person to learn specific knowledge and skills, which allows for skill development. The first step was conducted through an interview.

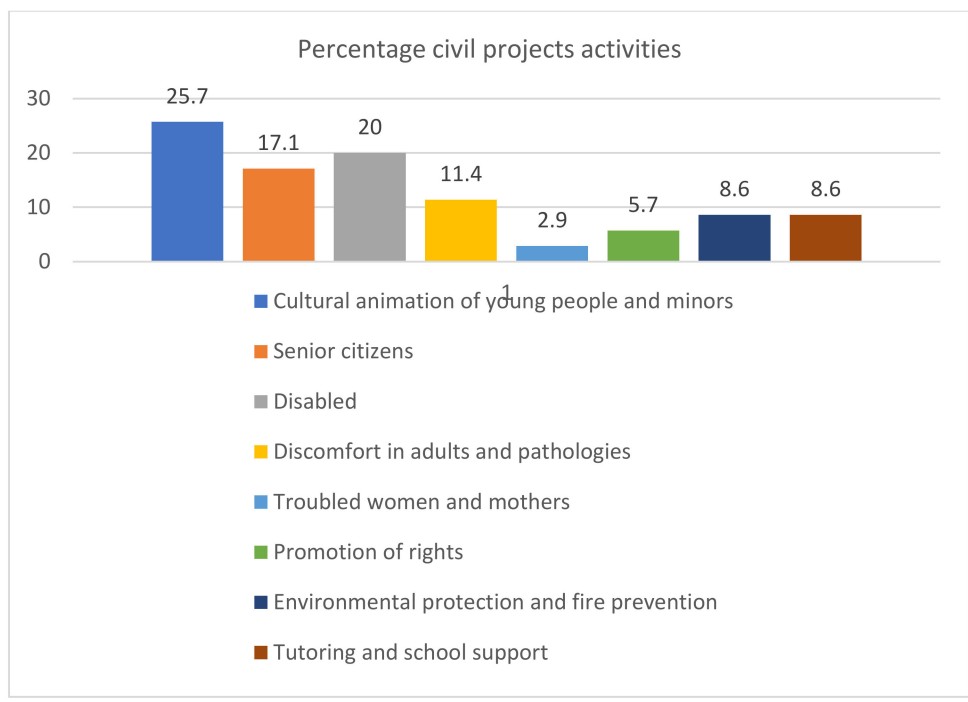

**Figure 2.** Percentage of Civial Project Activities.

In the second step, the subjects were helped by their tutors and had to reflect on what they have learned and the skills they developed during the year of Universal Civil Service. To understand which skills were acquired, the Lever Up tools were used, which allowed for a comparison between their level of competence and that which is described in the European standards. In the second step, the subjects were helped by their tutors. They had to reflect on what they learned and the skills they developed during their year of Universal Civil Service. The Lever Up tools were used to define the skills that were acquired, which allowed for comparison of their competence level with that described in the European standards. The third step involved documentation and collection of evidence on the skills acquired through various types of materials (photos, audio, video, reports, presentations, letters, storytelling). There was also an optional part in which it was possible to declare how the chosen skills related to school, work, volunteer work, or other experiences. The documentation and evidence collected were evaluated by an external commission, composed of Vol.To and the Faculty of Psychology, which analyzed the documents and evidence presented based on pre-established criteria and based on the level of complexity experience. In the last step, the subject acquired a greater awareness of his/her skills. The growth of awareness about these abilities will be useful to the volunteer to strengthen their self-esteem and for personal development. Finally, a certificate of validation of Lever Up skills will be issued, which can be used in various areas, such as work.

*4.3. Output*

The output included the 94 trained volunteers, the 42 h of generic training and the 75 h of specific training, the number of subjects who participated in the certification of skills and the project paths carried out.

*4.4. Outcomes*

The outcome achieved refers to participation in continuing training. The outcome is defined by the percentage of people who followed continuing vocational training after graduation. The outcome is related to the change in the skills and behavior of the subjects. Moreover, among the outcomes is also the impact of the number of hours of activities carried out thanks to the civil service in addition to those that the host subjects already carried out. Of the subjects in this study, 32.4% managed to validate at least three competences with appropriate evidence as required, while 44.1% preferred to focus exclusively on one of the 15 competences provided by the standards. The remainder did not complete the validation process.

*4.5. Social Impact Assessment*

The self-assessment associated with the certifications acquired highlights a first approach linked to personal change. At the end of the course, participants were provided with a questionnaire to map this aspect. The participants found a behavior change in almost all cases thanks to the skills and the path taken.

Furthermore, for 93.8% of participants, the change was good (21.5% good, 27.7% excellent, 44.6% excellent) while only 4.6% were considered sufficient and 1.5% insufficient. Of all the participants, 96.9% believe they increased their skills, are oriented to work on the project, and were able to identifythe soft skills that the project intended to instill in participants. The indicator is above the average as the 96.9% achieved by the Vol.To project exceeds the national average for the same parameter in the previous year by 15.8 percentage points. The certification of skills certifies the subject's ability. The second impact is linked to the activities carried out and generated 1440 h for each subject within the host institution, impacting the general population through activities of general interest.

The real novelty is being able to map the impact of the activities on the territory, which, through specific actions, respond to the public interest (or general interest) de facto in integrating state public activities.

*4.6. Interventionist Journey, Interview Results and Evaluating Volunteer Activity Approach*

The development and definition of the approach adopted for measuring the added value and the consequent social impact generated saw the collaboration of one of the authors with four key employees of the voluntary service center Vol.To who deal with deciding and developing the project and an official of the Italian council of Italian Council Presidency responsible for the project.

Based on previous experiences, the president of Vol.To had the need to identify and map the impact of civil service project. In particular, he declared, "I am sure the civil service has potential repercussions that must require an objective evaluation of the value generated given that the center co-finances the project." The need to map the impact and determine the added value is also stated by the official of the Italian Council Presidency who said, "We need to know if the objective of the project approved and shared by our offices has been achieved and what the results are based on to the public resources transferred." Duringvarious meetings, the two employees who deal with the definition of objectives, activities, and results highlighted two main aspects: "We need to identify the impact of training towards objectives of general interest and real change" and, "We understood that the activities in collaboration with the host third sector organizations have an impact on the whole system that allows to carry out activities that otherwise would not have been provided." No one has highlighted the difference between partner organizations participating in the project, different interests, or internal conflicts. With respect to the

specific question concerning the legitimacy of the hybrid organization configured in the project during the design phase of the approach to be adopted to evaluate the impact and relative value generated, the president of Vol.To states, "There are no problems with the legitimacy of collaboration between organizations with different objectives because they all converge towards a common goal that leads to greater welfare without creating real differences between all those involved." Indeed, all the subjects have taken for granted the sharing of the project's formal and informal objectives within a hybrid organization that sees the project presence of different subjects with opposite legal forms. All highlighted the importance of mapping the added value generated and determining the impact of the funding from a perspective of efficiency and analysis of the change generated, in this case by the subjects selected by the civil service.

The cost effectiveness and reliability of the measure are two fundamental requirements for measuring volunteering activities and confirm the national need for an international framework to measure volunteering activities [131]. The need to effectively identify and add value can be achieved through interviews with civil service volunteers. In agreement with all the subjects, the activities and changes, skills and abilities detected by each subject were evaluated. Some examples of interviews can be viewed at [125] but will be summarized in the outcome section. However, the method is subjective and not objective and cannot determine the enhancement of the volunteer activity's impact during the civil service on the system. The Italian Ministry of Labor and Social Policies already proposed a methodology to determine the value of volunteering in the third sector. The proposed scheme defines an indirect replacement method that allows identifying the hourly value of volunteering carried out to allow an objective quantification. During the periodic annual meetings, the interventionist approach made it possible to identify that the indirect method was the best one for determining the impact on context. According to the SIA analysis, all elements led to the quantification of the impact. The replacement cost can be determined through the following method:

$UWIFR = \sum t\ H_i V_i\ W_i$
UWIFR = replacement cost for single function
$H_i$ = average hours worked by volunteers in function i
$V_i$ = number of volunteers who performed the function i
$W_j$ = average salary applicable to function i

The approach identifies the real value of the volunteers' hourly activities based on the type of service carried out at the host organizations or at private or public companies that are partners of the initiatives. The determined value is shared within Vol.To and accepted by the Ministry which approved it as an objective value for determining the hours of volunteering. The determined value is shared within Vol.To and accepted by the Ministry which approved it as an objective value for determining the hours of volunteering.

### 4.7. Understanding Social Impact and Measuring New Value Creation

This section proposes the desired approach determining the added value linked to the specific activities carried out and the related social impact. The project's economic value starts from the evaluation of income and expenses related to the project. The revenues include public transfers linked to the recognition of expenses and revenues for consultancy to third sector entities that received the subjects in charge.

For the development of its social activity, Vol.To has generated economic value created which is distributed to human resources through the payment of wages and all related charges. Suppliers were remunerated due to the purchase of products and services necessary for the production of the services and management costs related to the structure. The value relating to personnel costs includes 68.40% of the activities carried out by the head of the Civil Service project in the figure of Dr. Maida Caria, for 17.63% of the activity carried out for the selection of participants in the course, for 12.02% linked to the activities of the internal trainer and a residual part for contracted external training. The head of the Civil Service's activities concerns the entertainment activity of institutional relations

with the region, the Department of Youth Policies, the university, the high schools and the Foreign Office of the City of Turin. The manager also managed the project sites' accreditation activities, planning, monitoring, research, selecting volunteers, maintaining relations with 35 tutors, and providing general training for 111 subjects (42 h for four classes). The activities of the head of the Civil Service lasted eight months.

The person in charge of the selection activities is configured with the management of institutional communications relating to the projects and participated in the selection interviews for a duration of two months. Simultaneously, the training phase that saw the specific design for the training course and delivery can be qualified with one month of activity.

Training activities related to 1.95% of the performance carried out by a safety expert (4 h module), activities of a communication expert (5 h module), and assistance for the legal part carried out by two lawyers (5 h).

The remuneration of suppliers, on the other hand, is divided by 96.94% into printing services for material used for the training course (roll-up printing, postcards, A3 posters, gift vouchers, plasticized signs for each organization, riders for each organization, A5 flyers, and printing personalized T-shirts for each volunteer) and the residual part for the hospitality offered to the Civil Service party. The management costs are completely related to expenses for the general training room used (42 h for four courses) and expenses for using the workstation equipped with PC (Office) for eight months.

The activities represented in the reclassification of financial items can also be assessed through a questionnaire administered to civil service volunteers at the end of the course. From the data obtained, another degree of satisfaction is highlighted concerning all the aspects evaluated. In analyzing the financial statement, there is no real distributed added value as the costs are higher than the project's revenue. Vol.To has covered the difference of €25,711 with its own operating income. In proportion, the project absorbed 1.32% of the total fund. Table 2 represents the financial statement of income and expenses.

**Table 2.** Prospect of income and expenses.

| Element | Values in € Year 2019 |
| --- | --- |
| Revenues from consultancy institutions | 9400.00 |
| Transfer for reimbursement of training expenses from the Department of Youth Policy–Italy Government | 7740.00 |
| **Economic value** | 17,140.00 |
| Remuneration of suppliers | 655.00 |
| Remuneration of personnel and utilities | 40,936.00 |
| Management fees | 1390.00 |
| **Economic value distributed** | 41,911.00 |
| **Economic value absorbed** | 25,711.00 |

From a first analysis, the civil service project absorbs rather than generates value, but this is wrong if one evaluates the impact and not the single reality. However, the volunteer activity carried out on the territory can be defined as the real distributed impact that did not represent financial terms. A total of 94 volunteers were employed, of which 36 were males and 58 females. In general, the qualification possessed by the majority of participants is in the Higher Middle School License (56 volunteers), followed in order by the University Diploma (14 volunteers), the Lower Middle School License (13 volunteers), and Degree (11 volunteers). Of 94 volunteers, 44 were employed in the city of Turin, and 48 in the Province of Turin.

Therefore, the impact of the civil service in economic terms includes the activity of 94 volunteers located throughout the region with a hypothetical remuneration of €17.11 per hour recognized in terms of enhancement of voluntary activities in the project. Volunteering

hours are recognized through the tabular criteria previous described (Section 4.2) associated with the fifth level of contributions (educator, hours without title, head worker, head cook, home assistant and tutelary services operator, hours of social assistance involved in basic assistance or otherwise defined, coordinator hours, teacher or manual and expressive activities, guidance with programming tasks, masseuse, entertainer, general nurse or childcare assistant with educational functions).

The same criteria were adopted by Vol.To on the occasion of the 2011 Single Deadline Call included in the financing activities of the projects of volunteer associations. The recognition and enhancement of voluntary work is a practice aimed at enhancing the third sector's contribution, methods, and push towards European recognition which took place by CSVnet in the national association of service centers for volunteering (CSV), which includes 62 of the 63 Service Centers for Volunteering.

Each volunteer worked 1440 h in one year. Therefore, the volunteers added value in the unrecognized territory for €24,638.40. Net economic contribution due monthly to each volunteer was equal to €439.50 per month and €5274 per year, which is equal to €19,364.40 for each volunteer. If the enhancement considers the number of active volunteers involved in the project, the value generated should be equal to €1,820,253.60. If we consider the value absorbed by Vol.To for public transfers and recognition by third sector entities, the net value is equal to €1,803,113.60. This value is further decreased by the economic value (net of transfers and contributions) absorbed for realizing the project (equal to €25,711), which brings the quantification of the activities carried out to €1,777,402.60. This is a value that does not find correspondence in financial result, but which is distributed throughout the territory. Therefore, out of 2500 Third Sector Entities accredited by Vol.To, the national civil service contributes €710.96 to the workforce for each. With 35 organizations involve in the project, the value for each organization of the hours worked would be equal to €50,782.93. According to the data, this activity has an impact on the provinces of Turin, Vercelli, Cuneo, and Asti. In 24 cases, there is expectation of a collaboration with public bodies such as civic libraries, hospitals, municipalities, schools, universities which would increase the overall value.

## 5. Discussion

The realization of the civil service through new organizational forms responds to the highlighted need to reduce available resources and obtaining a better impact in terms of performance [22]. The evidence provided by this case study analysis highlights how the social impact assessment and the breakdown into five phases of the analysis guarantees a real ability to determine and map the change up to the evaluation of the value generated through the quantification and qualification of social impact [25,26]. The process is innovative because SIA is applied together with interviews, interventionist analysis, and methodological determination of a hybrid organization's value. Vol.To is configured as hybrid organizations that directly carry out a coordination activity based on public funding and collaboration and by the host subjects (third sector, private and public partners) that involve the adhering subjects. The observation interventionist and the interviews confirm how hybrid organizations on specific welfare and social activities are able to overcome institutional limits with a capacity for dialogue between the different forms that strengthen the achievement of objectives [22,117,118]. Public funding that necessarily requires the involvement of other subjects in order to effectively respond to the need of the context for the allocation of voluntary resources of civil service on the territory on specific activities accompanies a vision of prolongation of interests and needs towards subjects with different forms, where business models and organizations share purposes and methods. The interviews highlight the shared need of the two financing organizations (Presidency of the Italian Council of Ministers and Vol.To) to define the territorial impact and value generated by highlighting an approach capable of mapping changes. In the literature, analyses of sustainability approaches and models focus on a single organization, proposing valid models exported to other types of participating organizations. In the case analyzed, the

approach analyzes cross-cutting elements such as social change, welfare, and educational activities carried out, and the economic context's impact, providing a broader and more generalizable vision. The analysis also highlights the elimination of the potential conflict between different interests usually identifiable in the network. The minimization of costs is evident for the public sector given the co-financing of Vol.To although this approach highlights from the accounting documents a greater control of inputs and outputs planned upstream and shared by all subjects with less bureaucratization of access to resources and the effective generation of value [11,14,16]. It is not possible from the analysis to state that the hybrid organization is better than direct management by the public sector. However, it is indisputable to affirm that the third sector's reform and the need to map the resources provided are due to an awareness of the limits already highlighted by the new public governance on the management and evaluation of performance between public and private entities. At the same time, the public sector alone would not be able to trace the market need, would not be able to identify the right placement of the subjects adhering to the civil service, and would not be able to provide proper training and assistance in the path of individual growth without stealing resources from a sector where austerity has already led to a reduction in staff in search of efficiency [30].

The orientation towards a common interest of public service leads to the determination of objective indicators capable of supporting and supporting the financing decisions and determining shared projects within hybrid organizations.

## 6. Conclusions

Through theory and case study, the study highlights that hybrid organizations generate social and economic value thanks to the change generated by the activities carried out. This value is determined in social–educational and economic well-being that would not have been successful without hybrid organizations. Therefore, the study confirms the first definition and configuration of generated value that is attributable to organizations oriented to the public interest [16,21]. The value generated is often an intangible value such as the change in skills or the economic fallout that does not present reporting by the organizations themselves. The case study represents for practitioners and academics an excellent starting case study to highlight all the cases in which public funding between subjects generates hybrid organizations for the principle of the prevalence of economic substance over legal form [118]. The social impact assessment adopted within the study provides a useful tool to map and determine the change given by the impact and the real value, which can be objectively quantified. The approach adopted can be generalized to any business and contributes at the literature on hybrid organization and social impact. Social impact assessment is a necessary tool to determine the value creation by hybrid organizations. In the case examined without the analysis, the hybrid organization had the determination of the outcome without being able to assess the real value creation that was determined on the change in individual skills subjects (social change) and the economic value distributed on the territory. Therefore, social impact assessment is one of the tools that public administrations and hybrid organizations must use to allocate resources and objectively enhance public interest activities inside the new complex organizations. The breakdown of impact assessment through the five elements of the value chain (inputs, activities, outputs, outcomes, and impact) guarantees a linear definition of the value generated through change with procedural objectivity capable of grasping hybrid organizations' complexity. The value generated or absorbed is the change generated by the impact measured based on the incidence of public resources allocated. The case study analyzed does not allow to generalize the metrics adopted for inputs, activities, outputs, results and impact. The impact that personalization requires is based on the context and objectives of interest and quantifying each country's added value [122,123]. The impossibility of generalizing the precise metrics but only the approach generates several research questions aimed at identifying indicators and measures that can be shared for each sector or social, environmental, and educational project. The study is part of the debate on evaluating

efficiency, effectiveness, and external context need of hybrid organization, providing new stimuli and reflections through the literature and case studies on value creation. Evaluating the performance, impact, and generated value of hybrid organizations initiates a debate within the area that is interested in post-modern public management. The analysis could begin a debate on the relationship between generated value and sustainability in the light of reducing expenditure and the response to epidemic crises.

Civil service impact assessment is common to many nations and requires a reproducible methodology for assessing change and generated value. The two approaches defined the impact in terms of skill change and individual approach with a certified change verification, and the real impact and economic value of the project concerning what was transferred and absorbed, which highlight real sustainability of the activity realized by the particular form of hybrid organization in the welfare system. The hybrid organization has an impact in social terms (for the change in the behavior of volunteers) and economic terms (for the value generated and the workforce made available to the third sector and private entities). It also increases the ability to respond to the questions of the market of social, welfare, and educational services envisaged by the project with the sharing of expectations, objectives, and cost-effectiveness required by the public sector, which is the project partner itself.

*Limitations and Future Research*

The study carried out presents only one case that should be analyzed in similar international contexts, although civil service projects highlighted characteristics and organizational change between adhering subjects are common. The approach adopted is suitable for analyzing the civil service's impact and value but could change metrics according to each hybrid organization's specificity. Future studies should highlight the impact of hybrid organizations and their generated value. When the literature is rich enough, it will be possible to determine common approaches to organizational clusters. The empirical evidence suggests that the approach could be guided by the common purpose of all the subjects that come together to achieve a common goal. This hypothesis should be refuted by new evidence.

**Author Contributions:** Conceptualization, P.E. and V.B.; methodology, P.E. and V.B.; validation, P.E.; formal analysis, V.B., C.F. and R.F.; investigation, V.B.; data curation, V.B.; writing—original draft preparation, V.B., C.F. and R.F.; writing—review and editing, V.B. and P.E.; visualization, V.B.; supervision, P.E.; All authors have read and agreed to the published version of the manuscript.

**Funding:** This research received no external funding.

**Institutional Review Board Statement:** Not applicable.

**Informed Consent Statement:** Informed consent was obtained from all subjects involved in the study.

**Data Availability Statement:** The data presented in this study are available on request from the authors.

**Conflicts of Interest:** The authors declare no conflict of interest.

## Appendix A

| Organization | Partner |
| --- | --- |
| A.I.B. Volontari Antincendi Boschivi-Sez. Bussoleno | A.S.D. ANIMA LIBERA: teaching of laboratories and courses A.C.F. MUSIC-ALL:singing lessons, show assembly. |
| A.S.D. L'isola che c'è | City of Druento: consortium of social assistance services of Pianezza-Day Center Management |
| Amici di Lazzaro | / |
| Amici di M.A.I.S. | M.A.I.S. Ong-CIFA Ong-RE.TE. Ong-Collaboration during workshops and in communication and promotion activities. Cascina Roccafranca e Associazione Variante Bunker: collaboration in the promotion and dissemination of materials, making available spaces for activities. |
| Area G Volontari | / |
| Associazione Don Bernanrdino Reinero | Associazione di Volontariato "Libro Aperto": collaboration for the creation and organization of events and workshops. |
| Auser Volontariato | Municipality of Turin and the police: transport network at affiliated hospitals and healthcare facilities. Cooperativa "La Valdocco": delivery of water to users reported by the cooperative and also organization of recreational activities in air-conditioned environments. City of Carmagnola-Transport and accompaniment of children with disabilities CISA 31-Transport to health facilities Municipal Pharmacy: free delivery of medicines for people over 75 or with walking problems. School complexes transport and accompaniment of pupils with handicap for playful support activities and psychomotor activities. |
| Banco Alimentare | / |
| Cascina Macondo | Associazione C.P: conducting the folk dance workshop-pedagogy and history of folk dance. Gruppo musicale Lumayna: use of the voice-popular song-rhythms-musical games. Municipality of Torino:search engine project-accreditation at the Ingenio Shop for the sale of artifacts and books; logistical support for the organization of events and dissemination of the project;contribution. VI circumscription di Torino: loan for the use of a small gym for the dance; theater workshop in Turin; contribution to the ceramics and storytelling workshops at the Via delle Querce facility where disabled people are accompanied by educators and OS of the Municipality |
| Contact | / |
| Cooperativa Quadrifoglio | Associazione Culturale Balancé: specific consultancy with its experts for the creation and monitoring of laboratories. |
| Cooperativa Quadrifoglio | Associazione Culturale Balancé: specific consultancy with its experts for the creation and monitoring of laboratories. |
| Cooperativa Quadrifoglio DUE | Associazione Culturale Balancé: specific consultancy with its experts for the creation and monitoring of laboratories. |
| Croce Rossa Susa | Cities: Piedmont Region: hospitals andreports; conventions and agreements as needed. |
| DI.A.PSI.- Difesa Ammalati Psichici-PIEMONTE | / |
| Don Bosco 2000 | / |
| Fondazione Circolo dei Lettori | Civic library of Torino: loan of books and reading aloud in hospitals, making instrumental and human resources available. Associazione La Brezza Onlus: involvement in the preparatory phase, through the provision of human resources. Host organizations: Humanitas Gradenigo hospital, Mauri-ziano Umberto I hospital, Molinette hospital, Univoc Institute, "Lorusso e Cutugno" prison. |
| Fondazione Piazza dei Mestieri Marco Andreoni | Associazione Piazza dei Mestieri: organization and management of project activities. Immaginazione e Lavoro soc.coop: promotion of activities proposed by the civil service. Cooperative La Piazza: graphic design and printing Vanni Editore-promotion and distribution, training of civil service volunteers. |

| Organization | Partner |
|---|---|
| Jeshua | Image Capture: supply of materials. La Piccola Casa della Divina Provvidenza, Cottolengo Torino: support at the project ü F.I.S.M. Torino, Centro di Formazione Francesco Faa di Bruno-Formazione Scuola dell'Infanzia: provision of material and human resources and spaces. |
| L'Altra Idea | Associazione Sportiva Dilettantistica HIDALGO Onlus: implementation of equestrian rehabilitation projects, collaboration with qualified therapists. |
| L'Isola che non c'è | / |
| La vita al centro | Associazione Biodanza Italia-Biodanza Teacher Training Baby Fox Association:provided professional entertainers. La Casa di Riposo San Vincenzo-The Cooperative Terra Terra, Azienda Agri-cola Apenocciola: provided an animator available for the creation and management of the G.A.S. |
| Legambiente Metropolitano | Istituto per l'Ambiente, Educazione Scholé Futuro Onlus: organization of workshops on air pollution and traffic control, environmental education activities in schools. |
| Merope | A.Ge. Piemonte: support for relationships with families. Ammp Sede Val Sangone: support in case of disabled transport. Echos Communication: support in the communication action. |
| Parrocchia Cottolengo | Ente Piccola Casa della Divina Provvidenza: to transmit these values to the minors and young people of the SGB Cottolengo oratory, through training meetings with operators and moments of sharing. |
| Presidio Sanitario San Camillo di Torino | University Hospital City of Health and Science Turin, School of Medicine of the University of Turin-Implementation of medical rehabilitation projects and research activities University of Turin, Department of Psychology: shared research activity. |
| S.E.A.–Servizio Emergenza Anziani | PRO NATURA Torino-Promotion of events, guided tours of the area. Ass. TELEHELP Turin-Te-rescue and telephone assistance. UNI.VO.C.A. Unione Volontari Culturali Associati: cultural events aimed at the elderly. ASSOCIAZIONE AMICI DELLA SACRA DI SAN MICHELE ONLUS: organization and provision of volunteer guides and witnesses for real and virtual visits. |
| S.E.A. delle Alpi | / |
| S.O.G.IT–Opera di Soccorso dell'Ordine di San Giovanni–I Giovanniti | S.O.G.I.T.-COMITATO REGIONALE PIEMONTE: activity coordination function. |
| Società Cooperativa Sociale La Mimosa | / |
| Sollievo | A.S.D. ANIMA LIBERA: teaching of dance workshops and courses. A.C.F. MUSIC-ALL: singing lessons, show production and assembly. |
| Specialmente Tu | / |
| VIVERE–Associazione volontari e famiglie con figli Portatori di Handicap | FONDAZIONE FRIMARIDE ONLUS: communication, awareness of the project. FORUM DEL VOLONTARIATO: search for volunteers and awareness of the project. |
| Vivi gli animali | Department of the Sustainable City, Municipality of Collegno: educational and awareness-raising programs, organization and promotion of information meetings and recycling laboratories, use of natural resources and recycling. San Donato Social Cooperative, Abele Lavoro Social Consortium: integration of people in socio-economic difficulties, educational and awareness-raising courses Djanet Association: educational and awareness programs. Tricycle Association: educational and awareness paths, use of natural resources and recycling. |
| Vol.To | Permanent Interregional Forum of Piedmont and Valle d'Aosta Volunteering:contacts with schools, project promotion, teacher meetings, scheduling of meetings, participation in the final event. S.A.A.-School of Business Administration of the University of Turin: provision of a communication expert for training. |

**Appendix B**

| Scope of Action | Type of Activity |
|---|---|
| Cultural animation of minors and young people | - Organization and management of workshops for children |
| | - Participation in the organization of events for minors and families |
| | - Sports activities management |
| | - Participation in the preparation of games, activities, sets |
| | - Planning of individual activities to be carried out with children and families |
| | - Organization of events and playful-recreational activities |
| | - Organization and management of proposals for local schools |
| | - Management of relations with external bodies |
| | - Secretarial activities: management of parental registrations and authorizations, reporting of events |
| | - Support for managing and maintaining contacts |
| | - Event communication through social media |
| | - Educational, linguistic, and relational support activities and participation in business workshops |
| | - Organization and participation in educational and recreational outings |
| | - Organization and participation in school camps |
| | - Supporting school educators/teachers in carrying out educational activities |
| | - Search through the web for initiatives aimed at children |
| | - After school support for school staff |
| | - Management and coordination of groups during the educational workshops |
| | - Support the Association's volunteers in information meetings in schools |
| | - Map the public places of aggregation of young people and organize meetings |
| | - Process data to improve the service |
| Senior citizens | - Social telephony |
| | - Accompanying the elderly |
| | - Animation and organization of playful activities |
| | - Support for socialization |
| | - Handling of paperwork with online forms |
| Disabled | - Listening to family members |
| | - Design of ad hoc activities for individual users |
| | - Organization and participation in individual and group outings with clients |
| | - Participation in apartment group activities |
| | - Management of creative workshops for the disabled (dance, poetry, reading, etc.) |

| Scope of Action | Type of Activity |
|---|---|
| Adult discomfort and pathologies | - Search for new food donors and personal contact |
| | - Monitoring of partnership requests from catering companies |
| | - Drafting of agreements with the charitable structures |
| | - Counter activities for collecting information, requests, needs |
| | - Reception and listening to people in conditions of hardship |
| | - Accompanying users in handling bureaucratic procedures |
| | - Assisted medical transport |
| | - Emergency services as stretcher bearers |
| | - Promotion of the culture of emergency through active participation in training opportunities for schools and citizens |
| | - Supporting hospital staff during targeted interviews with patients and their families in order to understand their needs. |
| | - Quarterly update of a report indicating the list of critical issues and patient needs |
| | - Research and analysis of the offer on the territory in terms of structures, facilities, associations and services in general that may be of help to the patient who returns to their home |
| | - Setting up moments of meeting between patients to encourage socialization and sharing their difficulties |
| | - Organization through the support of other professionals, of recreational activities |
| | - Contact discharged patients by telephone having feedback on the perceived well-being linked to the results of the rehabilitation process concluded in the structure. |
| | - Report through a specific report of the path developed during the project with the details for each patient taken in care |
| Troubled women and mothers | - Anti-trafficking prevention and support for former victims and single women |
| | - Support for entire families and very young people |
| Promotion of rights | - Research farms at Km. 0 |
| | - Management of the G.A.S. |
| | - Planning and realization of laboratories |
| | - Accompaniment in external activities |
| | - Realization of courses in schools (relationship with teachers and students) |
| | - Design of survey questionnaires |
| | - Planning of interventions against discrimination |
| | - Management of a class |
| | - Realization and management of multi-ethnic events with students |
| | - Mapping of intercultural events in Turin and its province |

| Scope of Action | Type of Activity |
|---|---|
| Environmental protection and fire prevention | - Monitoring of the territory and mapping of areas at risk of forest fire |
| | - Monitoring of the territory and sighting of outbreaks in periods of high risk of forest fires |
| | - Forest fire prevention activities with the cleaning of woods and forest areas |
| | - Information and awareness-raising activities for citizens and the public administration on the issues of preventing and fighting forest fires |
| | - Creation and updating of specific web pages |
| | - Implementation of training programs with the schools in the Municipality of Bussoleno |
| | - Organization of periodic network exercises between the public administration and the various associations |
| | - Promote workshops on new mobility to create opportunities for exchange between the various bodies |
| | - Stimulate the participation of citizens to dare new lifestyles and take care of air quality |
| | - Promote and disseminate good practices with information material, dossiers and statistical reports |
| | - Strengthen information and communication through the site, Facebook and YouTube |
| | - Meeting, knowledge and care of farm animals |
| | - Management of school visits and any hospitality |
| | - Use of natural resources and recycling |
| | - Care of the green and common areas |
| | - Organization of work groups for people in socio-economic difficulties |
| Tutoring and school support | - Scheduling of meetings in schools |
| | - Preparation of materials needed for the lesson, computer programs, exercises, textbooks. |
| | - Interviews with the teachers of the students followed alongside an educator |
| | - Cognitive meetings with families |
| | - Realization and administration of tests for the student |
| | - Data analysis |
| | - Study support activities |
| | - Monitoring of the activities carried out, the conduct and performance of students |
| | - Presence to guarantee children with SEN access to work in small groups |
| | - Presence in the after school and summer center to guarantee children with difficulties a positive experience in attending various activities |
| | - Presence in the canteen to allow all children to experience a situation of well-being during convivial moments with a child/adult ratio below the legal limits |
| | - Collaboration with the management and administration to organize cultural events |

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
