# Peer review of "Understanding Social Impact and Value Creation in Hybrid Organizations: The Case of Italian Civil Service"

_sustainability, doi:10.3390/su13074058_

Round 1

Reviewer 1 Report

The authors take up a very interesting research thread on understanding Social Impact and value creation in Hybrid Organizations. But they do so only partially. The article does not explain how the authors will  understand Social Impact specifically for hybrid organizations. The criteria for selecting this case study were also not explained. I have doubts whether this is really a hybrid organization. In my opinion, the article does not contribute anything essential for the science 

Author Response

The authors are grateful to the reviewer for highlighting a critical aspect regarding the significance and novelty of the selected case study.

The case study is significant and representative and hilights a novelty that can have many repercussions in the field of public management/hybrid organization and determination of the value generated. The case study is subject to public contributions and public control. Having to respond to reporting purposes of a public nature and public accountability, it is itself a hybrid organization due to the principle of the prevalence of economic substance over legal form [116]. In particular, the text of the sentence of the Court of Auditors of the Italian Ministry of Justice is highlighted, which recalls the principle of substance over form, effectively overcoming the mere aspect of financial transfer and determining the case study as a hybrid organization in fact, where money and the distribution linked to a specific project falls back and generates an organization whose goals, limits and internal interactions can be determined.

Refers to: SENTENZA N. 54 / A/2016 Corte dei Conti, nei giudizi di appello in materia di responsabilità amministrativa iscritti ai nn. 5329/A/Resp, 5322/A/Resp, 5344/Resp, promossi ad istanza  https://corteconti-iit.almawave.cloud/api/portal/downloadDocument/sentenze/SEZIONE%20DI%20APPELLO%20PER%20LA%20SICILIA/SENTENZA/54/2016

This aspect's significance lies in determining value and searching for practical and theoretical tools for determining value through new approaches that cannot be identified at the level of literature review.

Furthermore, the authors are grateful to the reviewer for pointing out a deficiency in the description of the definition of public value which has been referred to several times as a review in the text. Some significant parts are highlighted and the relationship and impact on hybrid organizations in the light of the case study:

The approach provided by hybrid organizations could support the correct use of re-sources, the limited use of the same response to a paradigm already studied by O'Flynn[14], who had questioned himself with the adoption of New Public Management in the twentieth century of managerial approaches borrowed from the private sector and applied to the public sector to increase public value. The public value generated by the adoption of new managerial approaches, as clarified by Stoker [15], includes the involvement of stakeholders in the collaborative and production process as an overcoming of the previous model; from this perspective, the analysis conducted sees the subjects traditionally involved as stakeholders on equal components of hybrid organizations.

O'Flynn [14] identifies in the concept of public value a new approach that adheres to that defined by hybrid organizations which, as theorized by the scholar, have a greater capacity for collecting preferences, a multiaccountability approach that involves all subjects, the ability to pursue multiple objectives, including service results, satisfaction, results, trust and legitimacy. The public value generated by hybrid organizations can be determined through the social and economic impact of the projects and services of public interest carried out [11,16]. Public value offers a broader way of measuring government performance and guiding policy decisions, according to Kelly, Mulgan, & Muers [17] public value could measure the impact of public interest projects. The concept of performance recalled in the paper therefore refers to the holistic conception of public value. The literature on hybrid organizations has not yet defined the public value it can generate either through a meaningful approach or case studies. The analysis conducted focuses on an example of a hybrid organization generated by a large project shared between the different types of organizations to merge them into a single organization with the same expectations and interests, and this is the Italian national civil service. The theoretical approach proposed to define public value in hybrid organizations can be generalized as it responds to the verifications and requirements identified by Ruddin [18], who also highlights how an approach that can also be generalized to other case studies has practical relevance in social studies. In the analysis conducted, the same hybrid organization approach applied to civil service projects can be adopted in different European contexts.

Reviewer 2 Report

Although the theme is interesting, I have major concerns regarding the article. Please see below my comments:

Abstract:

  1. The objective should be clear, namely regarding to what the authors mean by “measurement approach”. Is this related to change? The abstract does not provide evidence. In resume, and in my opinion, the paper’s research problem is not clear;
  2.  Contributions (both theoretical and practical) should be more clear (contributing to the debate is to ambiguous).

Introduction:

  1. The authors should adopt a suitable definition of “hybrid organization” regarding the papers’ context;
  2. See lines 52-57: suddenly, the theme changes: now the authors write about a survey. Who made the survey? Are the authors addressing the methodology? What phenomenon are the authors referring to? And what about performance´s measurement (what type of performance)? Please, be clear and objective. It is difficult for the reader to make any connection with the previous text, understand what is the research problem and what are the authors’ intentions. In fact, only after reading lines 201-204 I became more aware;
  3. Line 63: a case study can only be generalized in a theoretical stance (see, for example, Yin (2018);
  4. Lines 68-69: the authors are mixing methodology with data collection techniques. Furthermore, the type of interviews is not mentioned. Be more rigorous;
  5. See research questions. Starting these by the word “can”, the study is asking for a “yes or no” answer. Usually case studies seek, namely, the “how” and “why”;
  6. Lines 81-85: the authors do not mention a ”results“ or “finding” section;
  7. In my opinion, the introduction should be rewritten a more linear and focused way.

Theoretical framework:

  1. See lines 89-90: “analyze the academic debate on impact measurement”. Be clear. It is difficult to understand what the authors mean by debate on impact measurement;
  2. See lines 152-154: the sentence seems incomplete;
  3. Lines 169-174: see English. By using the term “while”, the sentences seem incomplete;
  4. Line 183: the tittle is somehow confusing;
  5. Lines 184-185: again, be more clear regarding to what do the authors mean by impact measurement;
  6. In line 194 the authors address Italy for the first time. In my opinion, in the introduction the authors should mention Italy when they write about the case study. Otherwise, why focusing on Italy in the literature review?
  7. See lines 208-209: the sentence “there are no studies on how to determine the impact
  8. of hybrid organizations objectively” is confusing. Which impact? Where?
  9. Line 226: the authors mention “public value”. There are several definitions of public value. Therefore, in my opinion the authors should address them and adopt a definition of “public value” for their study;
  10. The concept of performance is only addressed in line 265. In my opinion, it should be addressed earlier since it is related to the paper’ main goal. Also, the authors should adopt a definition for performance, one suitable to their study;
  11. General comment: the theoretical frame work should be written in a more “linear “away.

Method approach:

  1. In my opinion the authors should start this section describing the method (case study) and explaining why adopting such method. Furthermore, and again, be precise. For example, are we in the presence of a single, in-depth, exploratory case study? Writing “case study” is too broad. Possibly lines 289 to 331 can justify the use of the method.
  2. Data sources should be explicit. For example, depict the type of interviews and how many persons were interviewed. Were the interviews recorded and transcribed? Did the authors take notes? And what about observation? The authors should be clear regarding all sources.
  3. The authors should depict how they analysed such sources: by categorizing them into themes?
  4. Suggestion: restructure this section.

Some empirical evidence:

  1. In my opinion there is a lack of fit between the illustrations (which are mainly focused on training), the goal and the literature review;
  2. The authors should use more transcriptions in order to the reader better understand the case.

Discussion:

  1. There is no discussion towards the literature (no author was cited);
  2. Suggestion: By Merging the results’ section with the discussion, it would be easier for the reader to understand the case and its implications.

Conclusion:

  1. Line 734: again: a single case study only can be generalized theoretically;
  2. The authors should clearly state the main theoretical and managerial contributions of the study.

Annexes: Are they really necessary?

Author Response

Dear Reviewer,

Thank you for the punctuality of your comments and for the time you spent analyzing the paper.

Below, you'll find the timely response to your suggestions that have all been accepted by the authors.

The Authors

Abstract:

  1. The objective should be clear, namely regarding to what the authors mean by “measurement approach”. Is this related to change? The abstract does not provide evidence. In resume, and in my opinion, the paper’s research problem is not clear;
  2.  Contributions (both theoretical and practical) should be more clear (contributing to the debate is to ambiguous).

Answer:

  1. The objective is redefined as follow to highlight the problem:

The aim of the paper is twofold: first of all, it aims to analyze what kind of value is generated by hybrid organizations and how; then, it aims to understand the role of SIA (social impact assessment) in the measurement of added value, especially in terms of social and financial change generated by hybrids.

  1. The contribution is reviewed as follow:

After highlighting the gap in the literature, the study proposes an innovative approach, that com-bines SIA, interview, interventionist approach and documental analysis. Findings show that even if the approach can be generalized and contributes to the existing literature on hybrid organization and social impact, the metrics adopted for inputs, activities, outputs, results and impact should be adapted to the context and the objectives of interest. From a practical point of view, the study un-derlines the ability of hybrid organizations to respond to the requests of public services by sharing expectations, objectives, and cost-effectiveness with the public sector.

Introduction:

  1. The authors should adopt a suitable definition of “hybrid organization” regarding the papers’ context;
  2. See lines 52-57: suddenly, the theme changes: now the authors write about a survey. Who made the survey? Are the authors addressing the methodology? What phenomenon are the authors referring to? And what about performance´s measurement (what type of performance)? Please, be clear and objective. It is difficult for the reader to make any connection with the previous text, understand what is the research problem and what are the authors’ intentions. In fact, only after reading lines 201-204 I became more aware;
  3. Line 63: a case study can only be generalized in a theoretical stance (see, for example, Yin (2018);
  4. Lines 68-69: the authors are mixing methodology with data collection techniques. Furthermore, the type of interviews is not mentioned. Be more rigorous;
  5. See research questions. Starting these by the word “can”, the study is asking for a “yes or no” answer. Usually case studies seek, namely, the “how” and “why”;
  6. Lines 81-85: the authors do not mention a ”results“ or “finding” section;
  7. In my opinion, the introduction should be rewritten a more linear and focused way.

 Answer:

  1. The authors tried to define the concept of hybrid organization adapted to the case study and the reference context, as we refer to hybrid organizations as a model of integration between the public, social enterprises, cooperatives and the third sector. This aspect was mentioned already in the introduction and recalled precisely to distinguish it from the generic concept of Hybrid Organization provided by Billis (2010) which generically identifies hybrid organizations as integration of public, private and non-profit in a single new model in which the defects of each sector are surpassed by those of the other sectors. I refer to Billis, D. (Ed.). (2010). Hybrid organizations and the third sector: Challenges for practice, theory and policy. Macmillan International Higher Education. The authors already recall in the introduction:

Academics have been debating on the definition and outcome of hybrid organizations [1]. In the literature, there are three classic forms of hybrid organizations, hybrid pub-lic companies formed by subsidiaries and investee companies, social cooperatives to which the public provides a mandate for achieving specific objectives and social en-terprises [2,3]. However, the classical forms of hybrid organization are added to other more complex forms and have not yet been studied in the literature. An example is given by those third sector organizations that receive public funds, where in fact, the public sector is a partner in specific activities and which in some cases also involves the private sector to achieve shared social objectives. In these cases, the interest in the project's impact and the value created become recurring themes of interest for the new organizational forms created. The subjects involved in the hybrid organization pursue the common interest and support the achievement of social, environmental, or eco-nomic needs by overcoming potential obstacles thanks to the characteristics of each one who in such organizations unite to form a new organizational form [4].

  1. Considering lines 52 to 57 we agree on the lack of clarity of the first definition of performance and study. Therefore, the scholars have redefined and expanded the concepts by providing an in-depth definition of performance and public value recalled and analyzed.

The sentence:“The survey conducted intends to understand the phenomenon constituted by the new organizational forms and the associated performances' measurement.” Was not specific and correct and it is deleted.

The sentence is redefined from:

The concept of performance evaluation is associated with an impact or a change that generates public value [14].

To

The approach provided by hybrid organizations could support the correct use of resources, the limited use of the same response to a paradigm already studied by O'Flynn[14], who had questioned himself with the adoption of New Public Management in the twentieth century of managerial approaches borrowed from the private sector and applied to the public sector to increase public value. The public value generated by the adoption of new managerial approaches, as clarified by Stoker [15], includes the involvement of stakeholders in the collaborative and production process as an overcoming of the previous model; from this perspective, the analysis conducted sees the subjects traditionally involved as stakeholders on equal components of hybrid organizations. O'Flynn [14] identifies in the concept of public value a new approach that adheres to that defined by hybrid organizations which, as theorized by the scholar, have a greater capacity for collecting preferences, a multiaccountability approach that involves all subjects, the ability to pursue multiple objectives, including service results, satisfaction, results, trust and legitimacy.

The sentence is redefined from

This value is determined by the community's greater well-being, which, considering the social and service dimension, allows a real determination and enhancement of aspects that are sometimes intangible [11,15]. The literature on hybrid organizations to date does not study and identify neither the new form nor the impact generated, and the added value created in response to stakeholders' specific needs.

 To:

The public value generated by hybrid organizations can be determined through the social and economic impact of the projects and services of public interest carried out [11,16]. Public value offers a broader way of measuring government performance and guiding policy decisions, according to Kelly, Mulgan, & Muers [17] public value could measure the impact of public interest projects. The concept of performance recalled in the paper therefore refers to the holistic con-ception of public value. The literature on hybrid organizations has not yet defined the public value it can generate either through a meaningful approach or case studies.

  1. The reviewer advice “The case study can only be generalized in a theoretical stance”, he/she refers to “The case study can be generalized because it was carried out with the same characteristics in most European countries.”. It is changed in: “The theoretical approach proposed to define public value in hybrid organizations can be generalized as it responds to the verifications and requirements identified by Ruddin [18], who also highlights how an approach that can also be generalized to other case studies has practical relevance in social studies. In the analysis conducted, the same hybrid organization approach applied to civil service projects can be adopted in different European contexts.”
  2. The method adopted in the interview was recalled in the methodological section of the paper. The same approach was adopted with the adoption of the same method by:
    • Aleksandrov, E., Bourmistrov, A., & Grossi, G. (2018). Participatory budgeting as a form of dialogic accounting in Russia. Accounting, Auditing & Accountability Journal.
    • Secinaro, S., Brescia, V., Iannaci, D., & Jonathan, G. M. (2021). Does Citizen Involvement Feed on Digital Platforms?. International Journal of Public Administration, 1-18.

The research group thanks for the suggestions that led to the increase of the specific methodological section, which is not in-depth previously developed.

  1. Research questions were revised as suggested as follows:

RQ1: How do hybrid organizations generate value and what kind of value is it?

RQ2: How does the social impact assessment measure the phenomenon of value creation linked to change?

  1. The finding section in introduction is been added.
  2. The revisions requested allowed an improvement and a specificity useful to guarantee the understanding, placement and orientation of the study. The authors hope that the changes made respond to the need to focus more on the research topic and the analysis conducted.

Theoretical framework:

  1. See lines 89-90: “analyze the academic debate on impact measurement”. Be clear. It is difficult to understand what the authors mean by debate on impact measurement;
  2. See lines 152-154: the sentence seems incomplete;
  3. Lines 169-174: see English. By using the term “while”, the sentences seem incomplete;
  4. Line 183: the tittle is somehow confusing;
  5. Lines 184-185: again, be more clear regarding to what do the authors mean by impact measurement;
  6. In line 194 the authors address Italy for the first time. In my opinion, in the introduction the authors should mention Italy when they write about the case study. Otherwise, why focusing on Italy in the literature review?
  7. See lines 208-209: the sentence “there are no studies on how to determine the impact of hybrid organizations objectively” is confusing. Which impact? Where?
  8. Line 226: the authors mention “public value”. There are several definitions of public value. Therefore, in my opinion the authors should address them and adopt a definition of “public value” for their study;
  9. The concept of performance is only addressed in line 265. In my opinion, it should be addressed earlier since it is related to the paper’ main goal. Also, the authors should adopt a definition for performance, one suitable to their study;
  10. General comment: the theoretical frame work should be written in a more “linear “away.

 Answer:

  1. The section took up the topics covered in the literature review explaining the link. The authors did not want to clarify the text by recalling the debate on the topic of the special issue to which the paper responds: https://www.mdpi.com/journal/sustainability/special_issues/role_impact_assessment

The definition was clarified by recalling the call for paper:

The debate on the issue of social impact is generated by the consideration of multiple criteria of different nature (economic, environmental, and social), as well as the transparency and engagement of the different stakeholders, such as organizations, government, and communities oriented towards mapping resource sustainability in a complex environment.

  1. The line recalled was: “Other writers have gone further in a separate sector approach, looking at hybridization and hybrid organizations as the permanent features in the welfare system [47,48].”. It seems complete.
  2. Lines 169-174: while was replaced with instead. Thanks for the highlighting.
  3. line 187 point 2.3 the authors propose the new title: Hybrid organizations and the determination of the generated value
  4. The term impact measurement is defined as the reference definition: Gamble, E. N., Parker, S. C., & Moroz, P. W. (2019). Measuring the integration of social and environmental missions in hybrid organizations. Journal of Business Ethics, 1-14.

The sentence is integrated as follow: based on environment, workers, community, and governance and managerial approaches and frameworks adopted for measuring the previous elements. The economic aspect actually has an impact on the reference community and internal elements of the organization in the hybrid organization model [61].

  1. The choice of the authors is to introduce the reform in the theoretical part to recall the theme of public value and the relationship with the third and non-profit sector which has recently occurred in Italy and has already taken place in other countries. Reference is made to the text: Since 2017 in Italy, the country has witnessed the reorganization of the legislation on the third sector, including associations (social promotion association, voluntary associations), social cooperatives, social enterprises, ecclesiastical bodies, and foundations [64 ]. The new legislation increasingly directs existing organizations towards general interests of public interest, with objectives of social and environmental sustainability as already highlighted for the non-profit and the possibility of carrying out commercial activities with a type of taxation different from those profit companies communi-ty -oriented shared public purposes [65,66]. The third sector is increasingly configured as a hybrid organization without, however, having identified appropriate tools for measuring and evaluating the value generated by the impact of the activities often organized in collaboration with the public and private sectors.
  2. The two studies highlighted the absence in the existing literature reviews of a common approach that considers mission drift and challenges to legitimacy in hybrid organiza-tions, recalling particular social enterprises as a classic example of hybrid organiza-tion[41] and the difficulty in defining the value generated by hybrid organizations— public and third sector in the definition of value for the territory and the reference community [23].
  3. Recalling the O'Flynn and related studies, the definition of public value is recalled: The public value generated by hybrid organizations can be determined through the social and economic impact [11,16,74].
  4. The definition thanks to the suggestions and indications was recalled from the introductory objective by defining that the concept of performance is well defined by: Public value offers a broader way of measuring government performance and guiding policy decisions, according to Kelly, Mulgan, & Muers [17] public value could measure the impact of public interest projects. The concept of performance recalled in the paper therefore refers to the holistic conception of public value. The literature on hybrid or-ganizations has not yet defined the public value it can generate either through a meaningful approach or case studies.

Method approach:

  1. In my opinion the authors should start this section describing the method (case study) and explaining why adopting such method. Furthermore, and again, be precise. For example, are we in the presence of a single, in-depth, exploratory case study? Writing “case study” is too broad. Possibly lines 289 to 331 can justify the use of the method.
  2. Data sources should be explicit. For example, depict the type of interviews and how many persons were interviewed. Were the interviews recorded and transcribed? Did the authors take notes? And what about observation? The authors should be clear regarding all sources.
  3. The authors should depict how they analysed such sources: by categorizing them into themes?
  4. Suggestion: restructure this section.

Answer:

1.The longitudinal analysis of a case study allows, through different methodologies and approaches, to identify and explain the same phenomenon with different sources and visions, responding to the gaps that a single approach provides in literature; examples of longitudinal analysis of a case study are defined equally in the literature [117,118]. Please consider:

- Aleksandrov, E., Bourmistrov, A., & Grossi, G. (2018). Participatory budgeting as a form of dialogic accounting in Russia. Accounting, Auditing & Accountability Journal.

- Secinaro, S., Brescia, V., Iannaci, D., & Jonathan, G. M. (2021). Does Citizen Involvement Feed on Digital Platforms?. International Journal of Public Administration, 1-18.

The basic theory and approach that needs to be taken was published by Debois & Gadde in 2002 and sees 4921 adoptions. Dubois, A., & Gadde, L. E. (2002). Systematic combining: an abductive approach to case research. Journal of business research, 55 (7), 553-560.

  1. The interview is mentioned in: “2) creation of a chronological overview of the interview and interventions in the examined public value based on empirical data from documentation and interventionist workshops to structure a new approach to measure the impact and the social change. The interviews were carried out on the subjects who participated in the civil service at the end of 2020 and on the managers and employees of Vol.to between 2019 and 2020 to identify the best approach to be adopted for measurement. The interventionist approach involved the participation of one of the authors defining the tools, elements, and methodology to be adopted to measure social impact and value generated through an agreement between the University of Turin (Italy), Vol.to and the Presidency of the Italian Council of Ministers. The activity developed in 2020.”

At the same times the results of interviews, part of the SIA analysis are recalled in the subpoint “4.6 Interventionist journey, interview results and evaluating volunteer activity approach”. The interview at the volunteering different than the interviews at the hybrid organization subjects are recording at cite 121. Il Margine Storytelling - Le Testimonianze VOL.TO Versione 2; 2020 https://www.youtube.com/watch?v=DJgN1LcP0ks

 The interviews with the subjects involved in the civil service process were recorded entirely, some elements made up the storytelling present on youtube. More details on the approach adopted both for the interventionist recording the statements and for the results of the interviews were defined in the methodological section under point 2. It is redefining as follow:

2) creation of a chronological overview of the interview and interventions in the examined public value based on empirical data from documentation and interven-tionist workshops to structure a new approach to measure the impact and the social change. The interventionist approach was conducted through 5 meetings which took place from three months before the start of the project up to one month after the end of the project at the end of the reporting. The interviews were carried out on the subjects who participated in the civil service at the end of 2020 and on the managers and em-ployees of Vol.to between 2019 and 2020 to identify the best approach to be adopted for measurement. The interventionist approach involved the participation of one of the authors defining the tools, elements, and methodology to be adopted to measure social impact and value generated through an agreement between the University of Turin (Italy), Vol.to and the Presidency of the Italian Council of Ministers. The activity developed in 2020. The interviews with the volunteers, are developed by semi-structured questions aimed at defining the actual change provided by the project through the counterfactual method [121].

  1. The methodologies and the different sources converge towards the research questions addressed. The material and the approaches adopted for the analysis follow the structure of the papers published on journals present in the Journal Rankings and the ABS previously cited.
  2. The structure has the elements of the articles mentioned.

Some empirical evidence:

  1. In my opinion there is a lack of fit between the illustrations (which are mainly focused on training), the goal and the literature review;
  2. The authors should use more transcriptions in order to the reader better understand the case.

 Answer:

  1. The illustrations are based on the objectives defined by the same civil service project in most European countries. The analysis was agreed and conducted in order to determine the public value, as recalled social and economic. The environmental aspect is not applicable. The images and tables represent a transposition of the actual value through the five phases of the value chain and consequent SIA analysis as foreseen in the literature. The approach is the derivation of a practical and theoretical training project which provided for the certification of the skills on the social impact of the author who dealt with the interventionist analysis by the University and a European skills certification body. The approach on social impact assessment is one of those teachers and certified to one of the authors present in the SCH120 register held by CEPAS.
  2. As anticipated, the data collected are available online, references were provided in open access in order to guarantee the replication of the result by the other researchers. The determination of the social impact of the project described is part of an impact certification process by a European certifying third party. Furthermore, complete transcripts of all the interviews conducted and recorded are not as per the method and articles recalled, but only the significant extracts.

Discussion:

  1. There is no discussion towards the literature (no author was cited);
  2. Suggestion: By Merging the results’ section with the discussion, it would be easier for the reader to understand the case and its implications.

Answer:

  1. The appropriate references have been recalled as highlighted in the text.
  2. Previously the two sections were joined. But it was a stylistic choice mainly linked to the structure of the sustainability papers to divide the two sections.

Conclusion:

  1. Line 734: again: a single case study only can be generalized theoretically;
  2. The authors should clearly state the main theoretical and managerial contributions of the study.

 Answer

  1. The approach adopted can be generalized to any business and contributes at the literature on hybrid organization and social impact. However, the practical implication does not allow to generalize the metrics adopted for inputs, activities, outputs, results and impact. The impact that personalization requires is based on the context and objectives of interest and quantifying each country's added value [120,121].
  2. The study now highlights theoretical and practical aspects. Furthermore the case study represents for practitioners and academics an excellent starting case study to highlight all the cases in which public funding between subjects generates hybrid organizations for the principle of the prevalence of economic substance over legal form [116]. In particular, the text of the sentence of the Court of Auditors of the Italian Ministry of Justice is highlighted, which recalls the principle of substance over form, effectively overcoming the mere aspect of financial transfer and determining the case study as a hybrid organization in fact, where money and the distribution linked to a specific project falls back and generates an organization whose goals, limits and internal interactions can be determined.

Refers to: SENTENZA N. 54 / A/2016 Corte dei Conti, nei giudizi di appello in materia di responsabilità amministrativa iscritti ai nn. 5329/A/Resp, 5322/A/Resp, 5344/Resp, promossi ad istanza  https://corteconti-iit.almawave.cloud/api/portal/downloadDocument/sentenze/SEZIONE%20DI%20APPELLO%20PER%20LA%20SICILIA/SENTENZA/54/2016

Annexes: Are they really necessary?

The annexes allow a reader who does not know the reality to understand all the subtasks and dynamics that lead to consider the case study as a hybrid organization.

Round 2

Reviewer 1 Report

Well done, congratulations

Author Response

The authors are grateful for the suggestions and evidence that led to the increase of the paper. Currently he has had some insights requested by the second reviewer which have increased the clarity of some elements.

Reviewer 2 Report

The paper focus on an interesting theme and suffered an important improvement when compared with the first version. Even though, I still have some minor concerns regarding it, which the authors can see below:

Abstract:

  1. Regarding the theoretical contributions, the authors stress the importance of methodological aspects. In my opinion, the contributions should be aligned with the paper’s goal. The methodology is a means to such end.

Theoretical framework:

  1. Please review again previous lines 152-154, and specifically the following sentence, which seems incomplete: “They focused on the motivation of traditional entrepreneurs and, while some of them show the desirability for self-employment, tolerance for risk and self-efficacy at the center of their interests”. The problem may be on the word “while”
  2. Previous lines 169-174: see English. The sentence is still confusing. In my opinion the sentence “Vertical intra-organizational specialization indicates how formal authority is distributed among different levels of hierarchy, instead vertical inter-organizational specialization focuses on specialization among public organizations (ministries with many subordinate agencies)”. Could be split.
  3. Again, in previous line 194 the authors address Italy for the first time. The question remains. Why stressing the case of Italy when in their response to the reviewer, the authors claim that “The choice of the authors is to introduce the reform in the theoretical part to recall the theme of public value and the relationship with the third and non-profit sector which has recently occurred in Italy and has already taken place in other countries”. The authors could use this justification to stress the focus on Italy. In my opinion, the introduction would be the right section to do this. Otherwise, the authors do not provide a proper reason to highlight Italy in the literature review;
  4. The authors responded that “the data collected are available online, references were provided in open access in order to guarantee the replication of the result by the other researchers. The determination of the social impact of the project described is part of an impact certification process by a European certifying third party”. This should be explicit in the methodology section.

Conclusion:

In my opinion the conclusion is scarce regarding RQ2: How does the social impact assessment measure the phenomenon of value creation? (the conclusion should be more assertive regarding the “how” and RQ3: Is social impact assessment a useful tool to understand the use of public resources generated by hybrid organizations? Specifically, what are the paper’s conclusions about these topics?

Author Response

Dear Reviewer,

Thanks for the new comments and for taking the time to analyze the paper.

Below you will find the timely response to your suggestions that have been

accepted and integrated by the authors in the paper.

The authors

Comment

The paper focus on an interesting theme and suffered an important improvement when compared with the first version. Even though, I still have some minor concerns regarding it, which the authors can see below:

Abstract:

  1. Regarding the theoretical contributions, the authors stress the importance of methodological aspects. In my opinion, the contributions should be aligned with the paper’s goal. The methodology is a means to such end.

 Answer:

The findings of the abstract have been redefined to better highlight the answer to the research questions while preserving the innovation of the approach adopted. The abstract is currently composed as follows:

The aim of the paper is twofold: first of all, it aims to analyze what kind of value is generated by hybrid organizations and how; then, it aims to understand the role of SIA (social impact assessment) in the measurement of added value, especially in terms of social and economic change generated by hybrids. Hybrid organizations are a debated topic in literature and have different strengths in responding to needs, mainly in the public interest, nevertheless, there are not many studies that identify the impact and change generated by these organizations. After highlighting the gap in the literature, the study proposes an innovative approach, that combines SIA, interview, interventionist approach and documental analysis. The breakdown of SIA through the five elements of the value chain (inputs, activities, outputs, outcomes, and impact) guarantees a linear definition of the value generated through change with procedural objectivity capable of grasping hybrid organizations' complexity. The value generated or absorbed is the change generated by the impact measured based on the incidence of public resources allocated. Through the SIA and counterfactual approach, the civil service case study analysis highlights how the value generated by public resources can be measured; or otherwise more clearly displayed in the measurement process itself.

Theoretical framework:

 Reviewer:

  1. Please review again previous lines 152-154, and specifically the following sentence, which seems incomplete: “They focused on the motivation of traditional entrepreneurs and, while some of them show the desirability for self-employment, tolerance for risk and self-efficacy at the center of their interests”. The problem may be on the word “while”

Answer:

  1. The authors are again grateful for calling attention to the sentence. The sentence has been corrected as follows to give the correct meaning:

They focused on the motivation of traditional entrepreneurs; some of them show the desirability for self-employment, tolerance for risk and self-efficacy at the center of their interests [48]. 

Reviewer:

  1. Previous lines 169-174: see English. The sentence is still confusing. In my opinion the sentence “Vertical intra-organizational specialization indicates how formal authority is distributed among different levels of hierarchy, instead vertical inter-organizational specialization focuses on specialization among public organizations (ministries with many subordinate agencies)”. Could be split.

Answer:

The sentence was separated into two sentences providing, as suggested by the reviewer, a better understanding:

Vertical intra-organizational specialization indicates how formal authority is distrib-uted among different levels of hierarchy. Vertical inter-organizational specialization instead focuses on specialization among public organizations (ministries with many subordinate agencies).

  1. Again, in previous line 194 the authors address Italy for the first time. The question remains. Why stressing the case of Italy when in their response to the reviewer, the authors claim that “The choice of the authors is to introduce the reform in the theoretical part to recall the theme of public value and the relationship with the third and non-profit sector which has recently occurred in Italy and has already taken place in other countries”. The authors could use this justification to stress the focus on Italy. In my opinion, the introduction would be the right section to do this. Otherwise, the authors do not provide a proper reason to highlight Italy in the literature review;

Answer:

3.The section has been moved to the introduction highlighting both the Italian context and the impact of the reinforcement on the new hybrid organization models.

Reviewer:

  1. The authors responded that “the data collected are available online, references were provided in open access in order to guarantee the replication of the result by the other researchers. The determination of the social impact of the project described is part of an impact certification process by a European certifying third party”. This should be explicit in the methodology section.

 Answer:

4.The elements suggested and the process that saw the collection of information, the type of information and the certification of the collection from third parties were included in sub-point 3 of the methodology that combines SIA analysis and interventionist approach. The data used in open access on ZENODO were also cited and uploaded. Zenodo is the OpenAIRE project, in the vanguard of the open access and open data movements in Europe was commissioned by the EC to support their nascent Open Data policy by providing a catch-all repository for EC funded research. CERN, an OpenAIRE partner and pioneer in open source, open access and open data, provided this capability and Zenodo was launched in May 2013. In support of its research program CERN has developed tools for Big Data management and extended Digital Library capabilities for Open Data. Through Zenodo these Big Science tools could be effectively shared with the long-tail of research.

The section that was introduced is:

The data collected are available online, references were provided in open access in or-der to guarantee the replication of the result by the other researchers [125–127]. The determination of the social impact of the project described is part of an impact certification process by a European certifying third party. The approach is the derivation of a practical and theoretical training project which provided for the certification of the skills on the social impact of the author who dealt with the interventionist analysis by the University of Turin (Italy) and a European skills certification body – CEPAS [128,129]. The approach on social impact assessment is one of those teachers and certified to one of the authors present in the SCH120 register held by CEPAS.

Conclusion:

In my opinion the conclusion is scarce regarding RQ2: How does the social impact assessment measure the phenomenon of value creation? (the conclusion should be more assertive regarding the “how” and RQ3: Is social impact assessment a useful tool to understand the use of public resources generated by hybrid organizations? Specifically, what are the paper’s conclusions about these topics?

Answer:

The authors tried to further emphasize how social impact assessment can measure the value created and the social and economic value that allow the use of public resources:

Social impact assessment is a necessary tool to determine the value creation by hybrid organizations; in the case examined without the analysis, the hybrid organization had the determination of the outcome without being able to assess the real value creation that has been determined on the change in individual skills subjects (social change) and the economic value distributed on the territory. Therefore, social impact assessment is one of the tools that public administrations and hybrid organizations must use to allocate resources and objectively enhance public interest activities inside the new complex organizations. The breakdown of impact assessment through the five elements of the value chain (inputs, activities, outputs, outcomes, and impact) guarantees a linear definition of the value generated through change with procedural objectivity capable of grasping hybrid organizations' complexity. The value generated or absorbed is the change generated by the impact measured based on the incidence of public resources allocated.
